# A pilot study of closed-loop neuromodulation for treatment-resistant post-traumatic stress disorder

Jay L. Gill [1,2,16], Julia A. Schneiders [1,3,16], Matthias Stangl [1],
Zahra M. Aghajan[1,4], Mauricio Vallejo [1], Sonja Hiller [1], Uros Topalovic [1,5],
Cory S. Inman [6], Diane Villaroman[1], Ausaf Bari[4], Avishek Adhikari [7],
Vikram R. Rao [8], Michael S. Fanselow [1,7], Michelle G. Craske[1,7],
Scott E. Krahl[3,4], James W. Y. Chen[9,10], Merit Vick[11], Nicholas R. Hasulak[1,12],
Jonathan C. Kao[5], Ralph J. Koek[1,13], Nanthia Suthana[1,4,7,14,17] ✉ &
Jean-Philippe Langevin [4,15,17] ✉

The neurophysiological mechanisms in the human amygdala that underlie post-traumatic stress disorder (PTSD) remain poorly understood. In a first-of-its-kind pilot study, we recorded intracranial electroencephalographic data longitudinally (over one year) in two male individuals with amygdala electrodes implanted for the management of treatment-resistant PTSD (TR-PTSD) under clinical trial NCT04152993. To determine electrophysiological signatures related to emotionally aversive and clinically relevant states (trial primary endpoint), we characterized neural activity during unpleasant portions of three separate paradigms (negative emotional image viewing, listening to recordings of participant-specific trauma-related memories, and at-home-periods of symptom exacerbation). We found selective increases in amygdala theta (5–9 Hz) bandpower across all three negative experiences. Subsequent use of elevations in low-frequency amygdala bandpower as a trigger for closed-loop neuromodulation led to significant reductions in TR-PTSD symptoms (trial secondary endpoint) following one year of treatment as well as reductions in aversive-related amygdala theta activity. Altogether, our findings provide early evidence that elevated amygdala theta activity across a range of negative-related behavioral states may be a promising target for future closed-loop neuromodulation therapies in PTSD.

The biological mechanisms that enable threat detection are critical for survival but can contribute to post-traumatic stress disorder (PTSD) if overactivated following trauma. Identifying the neurophysiological correlates of these mechanisms in people with PTSD is critical for the development of personalized, neural circuit-based therapies. Thus far, functional neuroimaging studies in patients with PTSD have associated the affective response to trauma reminders with amygdala metabolic reactivity[1]. Rodent and human studies have demonstrated that fear-related memory retrieval and expression are associated with increased theta (~6 Hz) bandpower[2–7]. However, the neurophysiological correlates of emotional valence in the amygdala of patients with PTSD remain poorly understood. Further, how amygdala activity relates to clinically relevant behavioral states in individuals with PTSD are unknown due to the unavailability of intracranial recordings in affected individuals.

**Fig. 1 | Experimental protocol for iEEG recordings and clinical assessment.**
**a** Timeline of procedures used in TR-PTSD participants before (pre-stimulation
[Pre-Stim]) and after (post-stimulation [Post-Stim] 1-3) the onset of closed-loop
stimulation (Stim On). **b** Emotional Image Task showing example neutral, positive,
and negative images from the International Affective Picture System[10]. **c** Example
placement of a right amygdala electrode contact from a TR-PTSD participant, which
was localized using a post-implantation high-resolution co-registered CT
(top right).

In a rare opportunity, we recorded amygdala oscillatory activity
for one year in two individuals with treatment-resistant PTSD (TR-
PTSD) enrolled in an open-label clinical treatment trial (Clinical-
Trials.gov [NCT04152993]) using a responsive neurostimulation sys-
tem (RNS® System, NeuroPace Inc.). The RNS enabled intracranial
electroencephalographic (iEEG) recording during clinically relevant
states in the laboratory and at home[8,9]. Combined with longitudinal
tracking of PTSD symptoms (Fig. 1a), the current study (1) identified a
consistent and reliable amygdala neurophysiological signature of
negative emotional experience in two patients with TR-PTSD and (2)
tracked this effect and clinical responses to closed-loop stimulation
repeatedly over the full year.

## Results

### Aversive images increase amygdala theta activity in TR-PTSD

To characterize the oscillatory substrates of aversive experience,
amygdala iEEG activity was recorded during an Emotional Image Task
(Fig. 1a, b) in two TR-PTSD participants before stimulation onset (Pre-
Stim) and separately in six non-TR-PTSD participants implanted with
amygdala electrodes (Fig. 1c, Supplementary Figs 1–2) for seizure
monitoring and/or epilepsy treatment (Supplementary Tables 1–3).
During the Emotional Image Task, participants were shown photo-
graphs (scenes, objects, animals, people) selected from the Interna-
tional Affective Picture System (IAPS)[10]. Each image was categorized
as positive (valence > 6), negative (valence < 4), or neutral (4 ≤
valence ≤ 6) based on normalized valence ratings using a 9-point
scale[10] (Methods).

TR-PTSD participants showed a significant increase in amygdala
theta (5–9 Hz) bandpower 1–2 s following negative compared to
positive and neutral image presentation (Fig. 2a, b, Supplementary
Video 1; negative versus positive/neutral: model estimated mean dif-
ference [e.m.d.] ± s.e.m. = 0.27 ± 0.05, t(391) = 5.02, 95% Confidence

Interval (CI) = [0.17−0.38], $p < 0.001$, estimated effect size: Cohen's
$d = 0.49$ (Methods)). This negative-image-related amygdala theta
bandpower increase was absent in non-TR-PTSD participants (Fig. 2b,
Supplementary Fig. 3, Supplementary Fig. 4, negative versus positive/
neutral: model e.m.d. ± s.e.m. = −0.06 ± 0.03, t(701) = −1.95, 95% CI =
[−0.11 − 0.00], $p = 0.051$). Positive and neutral images were combined
due to a lack of a difference in theta power between these two con-
ditions (positive and neutral) in both groups separately (Methods).
Significant valence-related effects between negative vs. positive/neu-
tral images were not found in frequency bands other than theta
(5−9 Hz) in either group (Supplementary Fig. 3).

### Increased amygdala theta activity during trauma memories and symptoms

We next recorded amygdala iEEG activity while TR-PTSD participants
listened to recorded narratives of (1) their most traumatic experience
and (2) a pleasant memory during a Script-Driven Imagery Task[11]
(Fig. 1a (Pre-Stim), Methods). Amygdala theta power was elevated
during traumatic compared to pleasant memory exposure in both TR-
PTSD participants (Fig. 2c; traumatic versus pleasant: model e.m.d. ±
s.e.m. = 0.08 ± 0.03, t(30) = 2.57, 95% CI = [0.02−0.14], $p = 0.02$, esti-
mated effect size: Cohen's $d = 0.87$ (Methods)), suggesting that
amygdala theta activity encodes negative valence in both the visual
and auditory modality, and for both impersonal and personally rele-
vant stimuli.

We also analyzed iEEG recordings captured during self-reported
symptom exacerbations outside of the laboratory (Methods). During
these events, participants triggered the storage of iEEG activity (180 s/
recording) and provided a written description of symptoms and/or a
rating on the Subjective Units of Distress Scale (SUDS)[12] (Supplemen-
tary Table 4), for which we obtained 14 such recordings. Amygdala
theta bandpower was significantly increased in both TR-PTSD

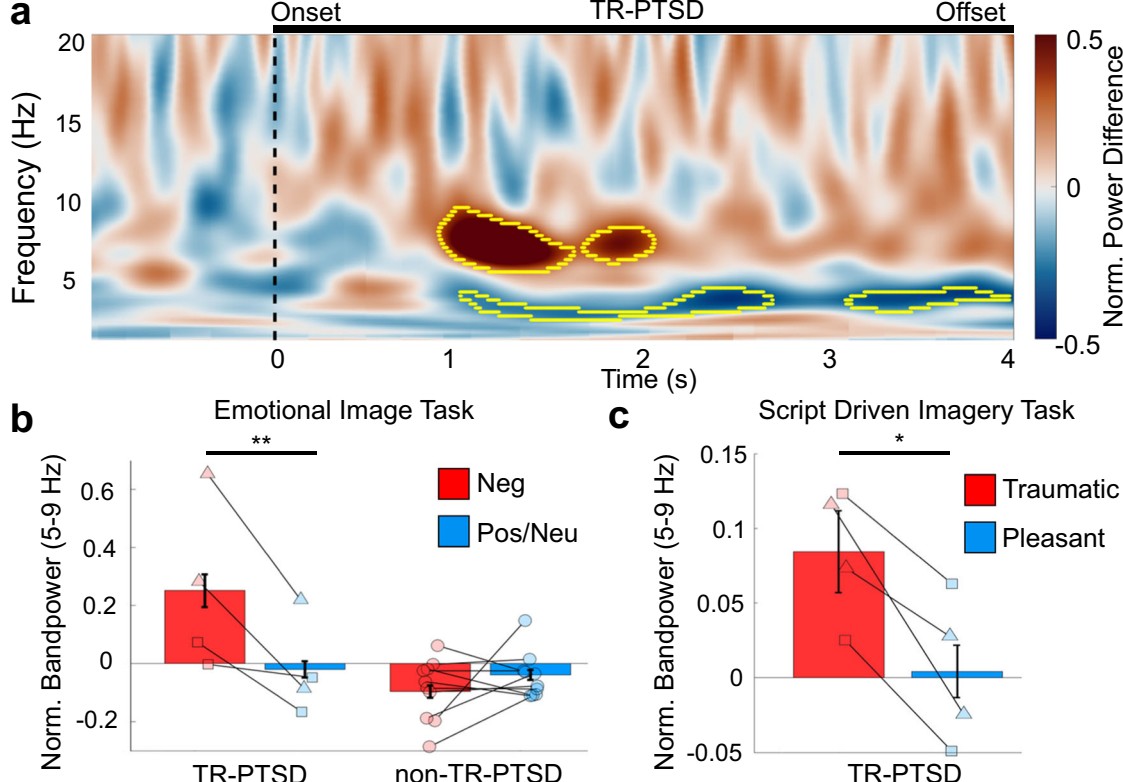

**Fig. 2 | Amygdala oscillatory encoding of negative valence and exposure to individual trauma reminders in TR-PTSD during pre-stimulation. a** Mean normalized (Norm.) difference in bandpower between negative versus positive and neutral (pos/neu) stimuli from the Emotional Image Task in TR-PTSD participants. Yellow outlined areas show ranges when trial type (negative or pos/neu) significantly predicted bandpower ($p < 0.01$, cluster-based permutation testing using a two-sided linear mixed model (Methods)). Black line indicates onset and termination of image presentation. **b** Norm. mean ± standard error of the mean (s.e.m.) amygdala theta (5–9 Hz) bandpower in TR-PTSD participants ($N_{participants} = 2$,

$N_{channels} = 4$ [left and right amygdala], squares = TR-PTSD 1, triangles = TR-PTSD 2) but not epilepsy (non-TR-PTSD, $N_{participants} = 6$, $N_{channels} = 9$) was significantly increased during the presentation of negative (red) compared to pos/neu (blue) images. ** = unadjusted $p < 0.001$ using a linear mixed model (Methods). **c** Norm. mean ± s.e.m. amygdala theta (5–9 Hz) bandpower in TR-PTSD participants ($N_{participants} = 2$, $N_{channels} = 4$, squares = TR-PTSD 1, triangles = TR-PTSD 2) during the presentation of individualized traumatic (red) and pleasant (blue) audio recordings (* = unadjusted $p = 0.02$ using a linear mixed model (Methods)).

participants during these symptomatic periods compared to scheduled recordings during participant-identified neutral periods (Supplementary Fig. 5, model e.m.d. ± s.e.m.: 0.17 ± 0.06, $t(244) = 2.70$, 95% CI = [0.05–0.29], $p = 0.007$). Notably, this effect was stronger in right hemispheric contacts in both participants (Supplementary Fig. 5b). Though larger studies are necessary to confirm potential lateralization of this effect and to examine if this is a key differentiation between aversive experience and symptom flares, this finding is consistent with prior studies that suggest a specific role for the right amygdala in PTSD[13,14].

**Closed-loop stimulation is related to theta activity and symptom reductions**
Given our findings of elevated amygdala theta activity during three different emotionally negative experiences (viewing of negative images, listening to traumatic audio scripts, and at-home symptom elevations), both TR-PTSD patients' RNS devices were programmed to deliver stimulation upon detection of sustained increases in low-frequency amygdala activity to mitigate symptoms (Supplementary Table 5). Notably, we found that the 5–9 Hz theta bandpower associated with aversive stimuli was also elevated during these stimulation-triggering periods (detections) compared to non-stimulation-triggering periods (non-detections) (Supplementary Fig. 6, Methods, theta bandpower during detections normalized to non-detections: TR-PTSD 1: mean ± s.e.m.: 0.05 ± 0.10, $t(1139) = 5.74$, 95%, CI = [0.04–0.09],

$p < 0.001$; TR-PTSD 2: mean ± s.e.m.: 0.06 ± 0.01, $t(789) = 4.83$, 95% CI = [0.04–0.09], $p < 0.001$).

Following 11 months of closed-loop stimulation, both TR-PTSD participants exhibited clinically meaningful reductions in PTSD symptoms, as measured by the Clinician-Administered PTSD Scale for DSM-5 (CAPS-5)[15] and the PTSD Checklist for DSM-5 (PCL-5)[16]. TR-PTSD 1 exhibited significant CAPS-5 reductions[17] (within-person change scores ≥ 13) during all clinical assessments following stimulation onset (month 2: 50.59% symptom improvement, month 3: 54.12%, month 4: 77.06%, month 5: 70.00%, month 6: 57.65%, month 7: 52.35%, month 8: 66.47%, month 9: 82.35%, month 10: 73.53%, month 11: 70.00%, month 12: 87.65%; (Fig. 3a, Supplementary Fig. 7a, c; subtests Supplementary Fig. 7e), while meaningful clinical improvements were observed in TR-PTSD 2 during months 4 (21.05% symptom improvement), 7 (32.11%), 8 (32.11%), 11 (40.00%) and 12 (36.84%) (Fig. 3b, Supplementary Fig. 7b, d; subtests Supplementary Fig. 7f). Subjectively rated PTSD symptom changes, as measured by PCL-5, were clinically reliable[17] (within-person change scores ≥15) during months 3–5 (month 3: 32.60% symptom improvement, month 4: 59.03%, month 5: 61.67%) and 7–12 (month 7: 29.96%, month 8: 47.14%, month 9: 57.71%. month 10: 55.07%, month 11: 56.39%, month 12: 63.00%) for TR-PTSD 1 and during months 11–12 (month 11: 34.03% symptom improvement, month 12: 27.75%) for TR-PTSD 2 (Supplementary Fig. 7a–d; subtests Supplementary Fig. 7g–h). Temporal differences in treatment response between TR-PTSD 1 and 2 may have been due to initial differences in the amount of stimulation

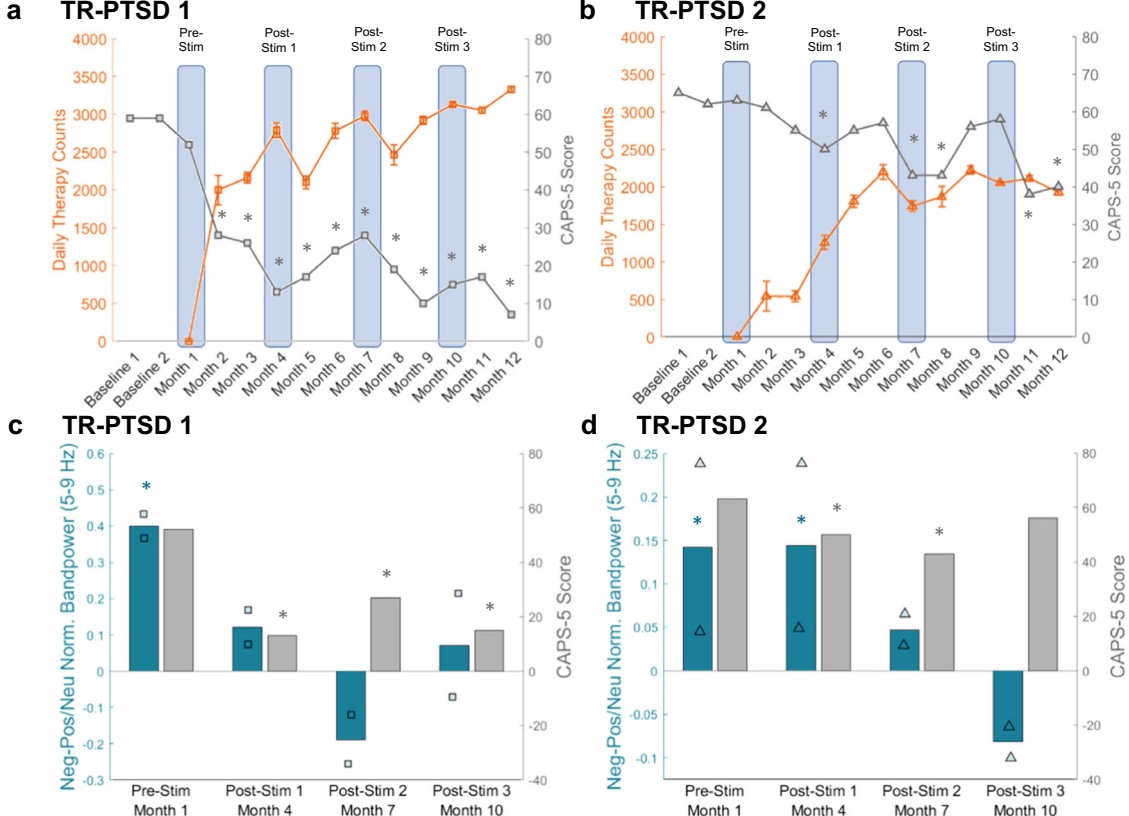

**Fig. 3 | CAPS-5 scores and negative-image-related amygdala theta activity during the Emotional Image Task decreased while daily stimulation therapy increased. a** Changes in CAPS-5 scores (gray) for TR-PTSD 1 during pre-stimulation (Pre-Stim) periods (baseline 1, 2, and month 1) and post-stimulation periods (Month 2–12) along with changes in mean ( ± s.e.m.) daily therapy (stimulation) counts (orange) over the same time period. Gray * = reliable change (Methods) in CAPS-5 scores according to Marx et al. in $n = 1$ participant[17]. relative to Pre-Stim CAPS-5 Scores. Blue boxes indicate time points when the Emotional Image Task was completed during Pre-Stim and Post-Stim sessions. **b** Same as (**a**) but for TR-PTSD 2.

**c** Difference in normalized (Norm.) amygdala theta (5–9 Hz) bandpower between negative (Neg) and positive/neutral (Pos/Neu) images from the Emotional Image Task (teal) and CAPS-5 scores (gray) for TR-PTSD 1 during Pre-Stim and Post-Stim sessions. Teal * = unadjusted $p < 0.001$, gray * = reliable change in CAPS-5 scores according to Marx et al.[17] relative to Pre-Stim CAPS-5 Scores. **d** Same as (**c**) but for TR-PTSD 2. Teal * = Pre-Stim unadjusted $p = 0.008$, Post-Stim 1 unadjusted $p = 0.038$, gray * = reliable change in CAPS-5 scores according to Marx et al.[17] relative to Pre-Stim CAPS-5 Scores. Squares = TR-PTSD 1, triangles = TR-PTSD 2.

delivered. Specifically, during the first three months of stimulation, TR-PTSD 1 received a greater frequency of stimulation than TR-PTSD 2 due to initial differences in their device programming (Fig. 3a, b, Supplementary Table 5, Supplementary Fig. 8). During the initial monitoring phase between surgery and stimulation onset, TR-PTSD 2 exhibited frequent bilateral low frequency elevations. Stimulation was thus only triggered if both the left and right amygdala showed increased low frequency activity to prevent potential overstimulation. This program resulted in a significantly lower frequency of stimulation in TR-PTSD 2 compared to TR-PTSD 1 (Supplementary Fig. 8). During month 4, stimulation parameters in TR-PTSD 2 were thus changed to match TR-PTSD 1, resulting in an increased number of daily stimulation therapies and subsequent clinical improvement in TR-PTSD 2 (Fig. 3b, Supplementary Fig. 8, TR-PTSD 2 stimulation therapies at Post-Stim 1 vs. Post-Stim 2: $W = 630$, $p < 0.001$).

Both TR-PTSD participants completed the Emotional Image Task again (with new images) at three time points after stimulation onset (Post-Stim 1-3) (Fig. 1a, Methods). During Post-Stim 1, TR-PTSD 1, but not TP-PTSD 2, exhibited both a reliable clinical improvement (CAPS-5 reduction) and an absence of negative-image related amygdala theta bandpower during the Emotional Image Task (Fig. 3c, d, Supplementary Fig. 9, negative versus positive/neutral: TR-PTSD 1 Pre-Stim: model e.m.d. ± s.e.m. = 0.40 ± 0.09, $t(196) = 4.29$, 95% CI = [0.22−0.58], $p < 0.001$; TR-PTSD 1 Post-Stim 1: model e.m.d. ± s.e.m. = 0.12 ± 0.11, $t(118) = 1.09$, 95% CI = [−0.10−0.34], $p = 0.28$; TR-PTSD 2 Pre-Stim:

model e.m.d. ± s.e.m. = 0.14 ± 0.05, $t(196) = 2.67$, 95% CI = [0.04−0.25], $p = 0.008$; TR-PTSD 2 Post-Stim 1: model e.m.d. ± s.e.m. = 0.14 ± 0.07, $t(118) = 2.10$, 95% CI = [0.01−0.28], $p = 0.04$. Interestingly, TR-PTSD 2 exhibited clinical improvement and reduction in negative-image related amygdala theta bandpower at Post-Stim 2 when the frequency of stimulation began to increase near the levels of TR-PTSD 1 (Fig. 3a−d, Supplementary Fig. 8, Supplementary Fig. 9b; negative versus positive/neutral: TR-PTSD 2 Pre-Stim: model e.m.d. ± s.e.m. = 0.14 ± 0.05, $t(314) = 2.55$, 95% CI = [0.03−0.24], $p = 0.01$; TR-PTSD 2 Post-Stim 2: model e.m.d. ± s.e.m. = 0.05 ± 0.09, $t(118) = 0.55$, 95% CI = [0.12−0.22], $p = 0.58$. This absence of amygdala theta bandpower differentiation between negative and positive/neutral stimuli was sustained at Post-Stim 3 even though there was no meaningful symptom improvement during this month (negative versus positive/neutral: TR-PTSD 2 Post-Stim 3: model e.m.d. ± s.e.m. = −0.08 ± 0.08, $t(118) = −1.07$, 95% CI = [−0.23−0.07], $p = 0.29$. Notably, TR-PTSD 1 maintained stimulation-related clinical improvement and an absence of negative image-related amygdala theta increases observed at Post-Stim 1 when assessments were repeated (negative versus positive/neutral: TR-PTSD 1 Post-Stim 2: model e.m.d. ± s.e.m. = −0.19 ± 0.10, $t(118) = −1.87$, 95% CI = [−0.39−0.01], $p = 0.06$; TR-PTSD1 Post-Stim 3: model e.m.d. ± s.e.m. = 0.07 ± 0.10, $t(118) = 0.74$, 95% CI = [−0.12−0.26], $p = 0.46$.

Similar post-stimulation-related changes in the amount of amygdala theta bandpower allocated to aversive stimuli were observed during repeated administration of the Script-Driven Imagery Task

(Methods, Supplementary Fig. 10a, Supplementary Fig. 11). Post-Stim sessions were combined due to the limited amount of data [4 trials of each condition/session] to compare traumatic and pleasant conditions (traumatic versus pleasant: TR-PTSD 1: model e.m.d. ± s.e.m. = 0.07 ± 0.03, $t(45) = 2.14$, 95% CI = [0.00–0.13], $p = 0.04$. TR-PTSD 2: model e.m.d. ± s.e.m. = 0.01 ± 0.04, $t(45) = 0.29$, 95% CI = [−0.06–0.08], $p = 0.77$). Furthermore, both participants reported decreased intensity of trauma memories triggered by the trauma narrative on the Responses to Script-Driven Imagery Scale (RSDI)[18] during the traumatic phase (Supplementary Fig. 10b, c, traumatic – pleasant RSDI score: TR-PTSD 1 Pre-Stim = 18.00, Post-Stim 1–3 = 6.00; TR-PTSD 2 Pre-Stim = 24.00, Post-Stim 1–3 = 18.67).

## Discussion

Altogether, these findings suggest that increased amygdala theta power is associated with processing of trauma-unrelated and related aversive stimuli, as well as symptomatic episodes in TR-PTSD. Using increased amygdala low-frequency power as a treatment target for closed-loop stimulation in TR-PTSD resulted in clinically significant amelioration of clinician- and patient-rated PTSD symptoms that were variable in one participant but very stable in the other. We also found reduced amygdala theta bandpower towards trauma-unrelated and related aversive stimuli following stimulation. Thus, amygdala theta activity could serve as both a treatment target and as a signature of therapy response in PTSD.

Prior investigations of iEEG activity during negative emotional stimuli processing in humans with epilepsy (without PTSD) report theta activity increases following the presentation of shock-associated neutral cues, but not following images of fearful faces[6,7]. Consistent with these findings, we do not observe significant image valence-related theta changes in our non-TR-PTSD (epilepsy) participants. A non-invasive functional magnetic resonance imaging (fMRI) study found that increased blood-oxygen-level-dependent (BOLD) activation in the amygdala occurred during the presentation of fearful faces but not negative valence images in healthy controls[19]. In patients with diverse stress-related psychiatric disorders, increased amygdala BOLD activation to fearful faces correlated with a memory bias towards negative stimuli[20]. Taken into consideration with our finding that negative-image-induced amygdala theta activity increased in TR-PTSD, but not in non-TR-PTSD, it is possible that amygdala theta reactivity is related to emotional salience, which may be altered in the setting of PTSD[21]. Future studies are needed to determine if heightened amygdala theta bandpower during negative images reflects pathological changes in amygdala circuitry, if these changes are specific to PTSD and/or other anxiety-related disorders, and if these effects are accompanied by changes in other physiological measures such as skin conductance response.

In line with other intracranial investigations in humans with epilepsy[6,7,22], we excluded data from seizure-onset zones and epileptic-related events (i.e., abnormal, sub-seizure, interictal epileptiform discharges [IEDs] from analysis). Though this minimizes potential differences in electrophysiological dynamics between groups, identification of a universal PTSD diagnostic biomarker is not feasible in the current investigation given the limited PTSD sample and epilepsy population. Thus, future studies across diverse patient populations (e.g., generalized anxiety disorder, obsessive-compulsive disorder, major depressive disorder) and larger samples of PTSD patients will be needed to determine whether heightened amygdala theta activity during aversive experience is ubiquitous across individuals with a PTSD diagnosis and to fully delineate its relationship with disease burden. Studies examining variations in symptom-related activation patterns across different traumas and sexes will also be critical to refine future closed-loop neurostimulation for trauma and sex-specific PTSD pathophysiology[23].

Functional neuroimaging studies have provided evidence for a correlation between amygdala hyperreactivity to emotional stimuli

and PTSD clinical severity[1]. Our current findings provide early evidence that amygdala theta power may similarly be related to PTSD symptomatology (i.e., greater theta reactivity may reflect increased symptoms). Specifically, we identify amygdala theta power as an ecologically valid, real-time signal that relates to self-reported symptom exacerbations in TR-PTSD, bridging decades of rodent work to the human clinical condition. Moreover, we have shown that amygdala closed-loop neurostimulation leads to a decline in both PTSD symptomatology and amygdala reactivity to aversive stimuli, which may be due to neuroplasticity-related changes over time. These findings are consistent with our previous case report of reduced PTSD symptoms after open-loop deep brain stimulation (DBS) of the amygdala[24] and of the subgenual cingulum and uncinate fasciculus (which links the prefrontal cortex and amygdala via fiber tracts)[25]. Though prior work demonstrating that amygdala emotional responses are fairly stable across repeated exposures[26] suggests that reduced theta reactivity shown here is likely related to chronic neuromodulation, and not habituation, future non-invasive and invasive neurophysiological recording studies aimed to track the effect of repeated emotional paradigm exposure will be needed to corroborate this hypothesis.

Recent studies using similar closed-loop or state-dependent neurostimulation approaches have shown improvements in major symptoms of Parkinson's disease[27], depression[28], binge eating[9] and obsessive-compulsive disorder[8,29]. Our results in TR-PTSD patients build on these studies and demonstrate that brief, intermittent closed-loop neurostimulation can be a promising method for treating chronic neurological and psychiatric conditions. Closed-loop stimulation provides several advantages including the ability to record electrophysiological signals over time, on-demand therapy delivery, and potentially extended battery life (and thereby fewer device replacement surgeries) depending on the programmed settings[30]. We demonstrate that the magnitude of change in theta power when encoding negative valence is sufficient to be detectable by commercially available closed-loop implantable neuromodulation systems initially designed to detect and treat seizures[31]. Since clinical improvements in this study were accompanied by reduced amygdala theta reactivity towards aversive stimuli, closed-loop stimulation may contribute to a normalization of altered brain circuity in PTSD. Though we show that theta activity is modulated by valence separately in two TR-PTSD participants across three separate behavioral paradigms, future studies using larger sample sizes will be needed to generalize our findings to the larger TR-PTSD population, determine scalability of approach, and further characterize network-related changes during encoding of negative emotional valence and/or re-experiencing of trauma in PTSD – namely, interactions between the amygdala and other potentially involved brain structures such as the prefrontal cortex, hippocampus, subgenual cingulum and uncinate fasiculus[1,25,30]. Additionally, though we do not observe image valence-related changes in non-TR-PTSD participants across recording modalities, further investigation will be needed to corroborate our findings and investigate potential differences in valence-related oscillatory changes across different amygdala subregions in individuals without TR-PTSD. Together, these findings are a first step towards the development of closed-loop neuromodulation strategies aiming to normalize emotional valence processing networks to improve PTSD symptoms.

## Methods

Reported research complies with protocol approved by the UCLA and VAHS Medical Institutional Review Boards (IRB) (UCLA: IRB #19-001203, VAHS: IRB #1615998). All participants provided written, informed consent prior to study enrollment. Participants did not receive monetary compensation in exchange for study enrollment. All participants were reimbursed for any costs associated with study participation.

## Participants

A total of 8 participants (all male, ages 28–68) with depth electrodes that were previously implanted for either treatment-resistant PTSD (TR-PTSD; $N_{participants} = 2$) or epilepsy (non-TR-PTSD; $N_{participants} = 6$) completed the study (Supplementary Table 1). TR-PTSD participants were diagnosed with severe, chronic combat PTSD (CAPS-5 score > 47 across > 8 weeks, total illness duration ≥5 years with no period of clinical remission), and clinically significant impairment in social and occupational functioning due to PTSD (≥70% service-connected disability, Global Assessment of Functioning Score (GAF)[32] ≤ 45, or no period of full-time employment for longer than 3 months in the past 5 years). Their PTSD symptoms corresponded to stage 2 of treatment-resistance for PTSD[33]. Clinical record documented failure to respond to adequate (minimum 3 month, with adherence) trials of at least 3 evidence-based treatments including at least one pharmacologic agent (sertraline, paroxetine, fluoxetine, or venlafaxine), and at least one trauma-focused individual cognitive-behavioral psychotherapy (either Prolonged Exposure Therapy [PE], Cognitive Processing Therapy [CPT], Eye movement Desensitization and Reprocessing [EMDR], or other form of evidence-based cognitive behavioral therapy for PTSD). TR-PTSD participants were chronically implanted with the RNS system (NeuroPace, Inc., Mountain View, CA) as part of a clinical trial (ClinicalTrials.gov [NCT04152993]), whereas epilepsy participants were implanted during clinical care, three with acute stereoelectroencephalography (sEEG) electrodes and three with a chronically implanted RNS System (NeuroPace, Inc., Mountain View) for evaluation/treatment of pharmaco-resistant epilepsy. Electrode placements of all participants were determined solely based on clinical treatment criteria (Fig. 1c, Supplementary Table 2, Supplementary Table 3). Participants were recruited from the Veteran Affairs Healthcare System (VAHS) and the Ronald Regan UCLA Medical Center (RRUMC). All participants provided written, informed consent according to a protocol approved by the UCLA and VAHS Medical Institutional Review Boards (IRB) (UCLA: IRB #19-001203, VAHS: IRB #1615998).

## Emotional Image Task

During this task, participants viewed images selected from the International Affective Picture System (IAPS)[10]. Images of varying emotional valence (positive, negative, neutral) were selected to match the number within each valence category and included similar numbers of scenes, objects, animals, and humans. Positive images had an average normative valence rating of 7 or above (1-most negative to 9-most positive). Average normative rating for negative images was below 4 and neutral images between 4 and 6. There was no significant difference in brightness, entropy, or contrast between the images of each valence (negative/neutral/positive) category (Supplementary Table 7)[34].

Task stimuli were delivered in pseudorandom order on a computer screen in sets of 30–33 images (60–99 images total). TR-PTSD 1 and 2 saw three sets of 33 images pre-stimulation and two sets of 30 images during each post-stimulation session. Pre-and post-stimulation image sets were matched in normalized arousal and valence for positive, negative and neutral categories (Supplementary Table 6). All other participants saw two sets of 30 images (60 total). All image sets had equal numbers of positive, negative and neutral images. Images were presented for a duration of 4 s. Participants were asked to provide self-paced subjective ratings (using a laptop trackpad) of valence, arousal and dominance immediately after each image presentation. After each rating, participants were instructed to fixate on a crosshair on the screen for 4 s after which the next image appeared. Normative ratings of objective valence from a sample of 100 participants[10] were used to separate negative, positive, and neutral image trials used in iEEG analyses. Positive and neutral images were combined due to a lack of a difference in theta power between these two conditions (positive and neutral) in both groups separately (TR-PTSD positive vs. neutral:

model e.m.d ± s.e.m. = 0.08 ± 0.05, $t(235) = 1.43$, 95% CI = [−0.03–0.18], $p = 0.16$; non-TR-PTSD positive vs. neutral: model e.m.d. ± s.e.m. = 0.03 ± 0.03, $t(467) = 0.97$, 95% CI = [−0.03–0.10], $p = 0.33$). We also compared image evoked activity to baseline to assess amygdala reactivity. In TR-PTSD, negative images evoked increased theta activity relative to baseline (TR-PTSD negative vs. baseline model e.m.d ± s.e.m. = 0.27 ± 0.04, $t(547) = 5.99$, 95% CI = [0.18–0.35], $p < 0.001$) while positive and neutral images did not (TR-PTSD neutral vs. baseline model e.m.d. ± s.e.m. = −0.04 ± 0.04, $t(514) = −1.14$, 95% CI = [−0.12–0.03], $p = 0.26$; TR-PTSD positive vs. baseline model e.m.d. ± s.e.m. = 0.03 ± 0.04, $t(514) = 0.84$, 95% CI = [−0.04–0.11], $p = 0.40$). In non-TR-PTSD negative and neutral images exhibited significantly different theta activity relative to baseline (non-TR-PTSD negative vs. baseline model e.m.d. ± s.e.m. = 0.036 ± 0.00, $t(861) = 4.19$, 95% CI = [0.02–0.05], $p < 0.001$; non-TR-PTSD neutral vs. baseline model e.m.d. ± s.e.m. = 0.03 ± 0.01, $t(855) = 2.49$, 95% CI = [0.01–0.06], $p = 0.01$). Positive images did not exhibit significant differences from baseline (non-TR-PTSD positive vs. baseline model e.m.d. ± s.e.m. = 0.007 ± 0.007, $t(868) = 0.99$, 95% CI = [−0.01–0.02], $p = 0.33$). During administration and analysis of the Emotional Image Task researchers were not blind to group (TR-PTSD vs. non-TR-PTSD group). The Emotional Image Task was assessed during months 1, 4, 7, and 10 of the clinical trial for both TR-PTSD participants and only one time for non-TR-PTSD participants.

## Script-Driven Imagery Task

Two participants with TR-PTSD completed a Script-Driven Imagery Task adapted from Pitman et al.[11], which included 60-s audio-recorded narratives (spoken by a male voice, present tense, second person) constructed from prior interviews about each patient's most impactful traumatic experience and a separate, pleasant experience, both containing visual, auditory, emotional, and cognitive information. Ratings of distress (0 not distressing at all, 100 extremely distressing) of each script by 5 blinded raters (3 female, mean age = 30.2 years) were significantly lower for the pleasant compared with traumatic scripts for each participant (TR-PTSD 1: pleasant: mean ± s.e.m. = 4.00 ± 2.26, traumatic: mean ± s.e.m. = 88.00 ± 4.79, $t(4) = 14.12$, $p < 0.001$, 95% CI = [2.08–10.62]; TR-PTSD 2: pleasant: mean ± s.e.m. = 17.40 ± 8.53, traumatic: mean ± s.e.m. = 73.60 ± 5.19, $t(4) = 13.09$, $p < 0.001$, 95% CI = [1.92–9.85]). Each of the pleasant and traumatic imagery script trials were repeated 4 times, in a fixed order (pleasant followed by traumatic) to prevent anxiety elicited by the traumatic script from persisting during the pleasant script, as previously described by Bremner et al.[35]. For each trial, each participant listened to the script for 60 s, imagined the scene for another 60 s with eyes closed, followed by a 30-s recovery period. After each block of pleasant and traumatic trials, participants rated their acute PTSD symptoms that they experienced during the previous exposure on the Responses to Script-Driven Imagery Scale (RSDI)[18]. The Script-Driven Imagery Task was assessed at the beginning of month 2 (Pre-Stim), and during months 6, 8, and 12 (Post-Stim 1-3) for TR-PTSD 1 and at the beginning of month 2 (Pre-Stim), and during months 4, 8 and 12 (Post-Stim 1-3) for TR-PTSD 2.

## Tracking PTSD symptom severity

PTSD symptom severity was tracked in both TR-PTSD participants using the Clinician-Administered PTSD Scale for DMS-5 (CAPS-5)[15] and the PTSD Checklist for DSM-5 (PCL-5)[16]. During each Clinical Assessment visit, participants rated their PTSD symptom severity on the PCL-5 questionnaire. CAPS-5 assessment interviews over the past month were administered and rated by the study psychiatrist (R.K.) and clinical psychologist (J.S.). CAPS-5 and PCL-5 scores were collected 5–6 months and 1 month before surgery (baseline 1 and 2), 1 month after surgery (before stimulation initiation) and during months 2–12 post-stimulation for a total of 14 measures in each TR-PTSD participant.

To assess the magnitude of individual PTSD symptom change due to stimulation, we computed difference scores of mean CAPS-5 and PCL-5 scores across assessments before (baseline 1 and 2, month 1) and each assessment after stimulation onset (months 2–12). Symptom changes for months 2–12 for CAPS-5 and PCL-5 were classified as indicative of reliable change when the difference scores were greater than or equal to the respective threshold (13 for CAPS-5 and 15 for PCL-5) for identifying clinically meaningful change for male combat veterans (sample 1) as reported by Marx et al.[17]. We calculated % symptom improvement on the CAPS-5 and on the PCL-5 for each participant by using the following formula: % symptom improvement = [(pre-stimulation baseline score (mean of baseline 1, baseline 2, month 1) – score of each month after stimulation onset (post-stim)) / pre-stimulation baseline score] × 100.

## Tracking daily therapy counts
In both TR-PTSD participants, the number of stimulations that each participant received per day as triggered upon the programmed detections by the RNS device was stored in the NeuroPace Patient Data Management System (PDMS).

## Electrophysiological data acquisition
Intracranial electroencephalographic (iEEG) data was recorded using either the chronically implanted NeuroPace RNS System or acute depth electrodes that were implanted for sEEG. Further details are below.

**RNS.** Five participants had a NeuroPace RNS System neurostimulator and leads already implanted for at least 2 months prior to participation in the research study. The neurostimulator and leads were implanted according to the Food and Drug Administration (FDA) approved indication for use in three participants with drug-resistant focal-onset epilepsy and programmed to detect abnormal electrical activity in the brain and respond in a closed-loop manner by delivering imperceptible levels of electrical stimulation intended to reduce seizures. In two participants, the neurostimulator and leads were implanted according to this investigational device exemption (IDE) to investigate responsive neurostimulation to treat TR-PTSD (Clinicaltrials.gov NCT04152993). Each participant provided informed consent, according to an approved Institutional Review Board (IRB) protocol, to temporarily turn off stimulation during the experimental tasks to avoid stimulation artifacts in the recorded iEEG data.

Each neurostimulator was connected to 2 leads that each contained 4 macro-contacts. Contacts were spaced 3.5 mm apart and bipolar iEEG activity (i.e., where the signal reflects differential activity between two neighboring contacts) was recorded with a sampling frequency of 250 Hz. Participants were implanted with either the RNS-300M or RNS-320 neurostimulator model (Supplementary Table 1). Prior to data collection, amplifier settings were changed from clinical default settings to a 4–120 Hz filter with −3 DB attenuation for model RNS-300M, and a 1 Hz high-pass and 90 Hz low-pass filter for model RNS-320. Despite the 4 Hz high-pass filter in the 300 M model, frequencies between 3 and 4 Hz were included in analyses similar to previous studies[22,36], which show that the filter's frequency response at 3–4 Hz frequencies show attenuation of only about 20%[36]. Moreover, in all plots and analyses that utilize frequency power, individual frequency steps were normalized relative to the channel's activity for each frequency step (i.e., 3 Hz power at any given time point is z-scored relative to the 3 Hz power over the entire experimental recording period), which further accounts for the band-pass filter's attenuation of amplitudes between 3-4 Hz in model RNS-300M, and allows a comparison of frequencies independent of their overall amplitude.

During the experimental tasks, iEEG data were continuously monitored and recorded using a NeuroPace Near Field Telemetry Wand that was secured over the neurostimulator implant site on the participant's head. The experimental setup did not involve a wired connection between the recording device (the implanted neuro-stimulator) and an external power source, making signal contamination with line noise unlikely. The experimental setup, including synchronization methods have been described previously[22,37].

**sEEG.** Depth electrodes (Adtech, Racine, WI) were implanted stereo-tactically using brain magnetic resonance imaging (MRI) and digital subtraction angiography (DSA) guidance. They included platinum contacts for iEEG recording with each electrode containing eight 1.5-mm-wide macro-contacts for clinical use. Each electrode contact was connected to an electroencephalogram amplifier (Nihon Kohden) and recorded at variable sampling rates (non-TR-PTSD 2: 200 Hz, non-TR-PTSD 1, non-TR-PTSD 3: 2 kHz), saved and exported as European Data Format for further analysis with MATLAB (Mathworks, Inc., Natick, MA). For communication between the iEEG recording system and the laptop displaying the Emotional Image Task, synchronization pulses were sent via AUX output. Following data collection, recordings were converted to a bipolar montage to match the RNS recording configuration. Bipolar montages were created by taking the difference in activity between adjacent contacts localized in the amygdala.

## Remote iEEG recording during TR-PTSD symptom exacerbation
To assess electrophysiological changes present during TR-PTSD symptom exacerbation and asymptomatic periods in real-world settings, we utilized the remote recording capabilities of the RNS device. During periods of self-reported symptom exacerbations over one year, TR-PTSD participants were able to wirelessly trigger the storage of iEEG activity by swiping a magnet over the scalp at the implant site. Proximity between the implanted neurostimulator and the magnet enables the storage of an iEEG trace 120 s before (pre) and 60 s after (post) the swipe. We obtained 3 recordings for TR-PTSD 1 and 11 recordings for TR-PTSD 2. Following the magnet swipe, participants indicated the intensity of the emotional event using the Subjective Units of Distress Scale (SUDS)[12] and/or self-reported descriptions. For TR-PTSD 1 there were two self-reported descriptions of symptomatic events, one of them was rated on the SUDS scale with a score of 8 out of 10 (Supplementary Table 4, event 1). Magnet swipes for TR-PTSD 2 ($n = 11$) had a SUDS score of 2.23 ± 0.33 (mean ± s.e.m.) out of 10. iEEG recordings during asymptomatic periods were collected using scheduled recordings where RNS devices were individually programmed to record iEEG activity during times of the day when participants tended to report being asymptomatic. These asymptomatic time periods occurred between 8 am and 1 pm (TR-PTSD 1) and 10–11 pm (TR-PTSD 2).

## Electrode localization
The precise anatomical location of each electrode contact was determined by co-registering a high-resolution post-operative head computed tomography (CT) image (1 s rotation, high-quality (HQ) mode, helical pitch 1, 0.6 mm slice collimation) to a pre-operative high-resolution structural MRI image (T1-weighted sequence, TR-PTSD 1-2, non-TR-PTSD 1-4: TR = 2100 ms, TE = 2.84 ms, 192 slices, voxel size: $1.0 \times 1.0 \times 1.0$ mm$^3$; non-TR-PTSD 5-6: TR = 1900 ms, TE = 2.44 ms, voxel size: $1.0 \times 1.0 \times 1.0$ mm$^3$) for each participant. Electrode localizations (Fig. 1c, Supplementary Fig. 1, Supplementary Fig. 2, Supplementary Table 2, Supplementary Table 3) were confirmed via 1) automatic segmentation using the FreeSurfer software suite[38] and 2) visual inspection by trained lab personnel (J.G., N.S., J.P.L.).

## Detection of epileptic events
To detect interictal epileptiform discharges (IEDs) in epilepsy participants, for each iEEG channel we applied a method described previously[39], and which we have used in previous studies with the RNS System[22,36,37]. This method uses a double thresholding algorithm with

two criteria: (1) the envelope of the unfiltered signal was 6 standard deviations above baseline; or (2) the envelope of the filtered signal (15-80 Hz band-pass filter after signal rectification) was 6 standard deviations above baseline. Trials where there was an IED detected 2 s before or during image presentation were excluded from analysis. Though the total IED detected represented 2–4% of the total iEEG signal across non-TR-PTSD participants, an average of $5 \pm 0.826$ of 60 trials were excluded from each electrode. No channels involving neurologist-identified seizure foci were incorporated in analyses. All non-TR-PTSD (epilepsy) participants gave informed consent to turn off the detection of epileptic events and consequent stimulation during completion of the experimental paradigm to prevent stimulation artifacts in the iEEG recordings.

### Electrophysiological analyses

Intracranial EEG activity was visually inspected for 60 Hz noise using time frequency power spectrograms generated using the BOSC (better oscillation detection) toolbox. Additionally, iEEG trials underwent epileptic event analysis for exclusion of trials with aberrant activity (*Detection of Epileptic Events*). The BOSC toolbox was used to calculate instantaneous power[40]. We performed complex Morlet wavelet convolution to calculate power spectra for all recordings, across a frequency range of 1–125 Hz in steps of 0.25 Hz[40,41]. For each frequency step, a 6th order (6-cycle) complex Morlet wavelet, stretched in time, was created and used as the kernel for convolution with the iEEG signal. The result of convolutions were complexed-value signals from which we could extract phase, amplitude, and power to calculate time-frequency spectrograms and power spectrum density (PSD). To ensure an equal number of cycles for each frequency, we defined a fixed width (6th order) for the complex Morlet wavelet based on previously published reports, which also used wavelet transforms for time-frequency analyses[39,40]. A 6th order wavelet was used to optimize the tradeoff between time and frequency resolution where an increased width (number of cycles) of the wavelet gives reduced estimation of rapid frequency changes (reduced temporal specificity) while the range of frequencies that can be estimated is increased and vice versa. In the initial analysis, to locate the low frequency band of interest, the list of frequencies went from 1 to 20 Hz in 0.25 Hz increments. Following identification of 5–9 Hz as being significantly modulated by valence in TR-PTSD, the list included frequencies from 5 to 9 Hz in 0.25 Hz segments.

### Linear mixed-effect model analysis

Frequency and temporal features of interest were identified using a linear mixed effects model (LMM) implemented using the fitlme function in MATLAB. Within the LMM, participant and electrode contact were treated as random effects. The restricted maximum likelihood method was used to estimate an effect of type (negative or positive/neutral). In this way, the model determined whether the estimated difference in frequency power between negative and positive/neutral (treated the same) images was significantly different from zero. We performed this strategy at each time (0–4 s following image presentation) and frequency (0–20 Hz) step across channels in TR-PTSD participants to empirically identify the time and frequency ranges where low-frequency activity differed between negative and positive/neutral images. Significant temporal and frequency ranges were detected using cluster-based permutation testing. During this procedure, a null time-frequency power spectrogram was created by shuffling condition labels (negative vs. positive/neutral). An LMM was performed at each time-frequency power pair and a *p* value of <0.05 was used to determine significance. Clusters (consecutive time-frequency pairs of significance) were then detected, and the size of the largest cluster was saved. This procedure was then repeated 1000 times. Cluster sizes derived from the observed (unshuffled data) that were larger than the 99th percentile of the shuffled distribution were labeled as significant.

LMMs that were used to assess the effect of average bandpower within the frequency and time points of interest across negative and positive/neutral conditions were implemented using each participant's theta (5–9 Hz) bandpower (normalized to theta bandpower over the entire session) from 1–2 s following image presentation as a 'response variable' and image type (positive, negative, neutral) to predict oscillatory bandpower variability in response to image presentation. All predictor variables were specified as fixed effects. Participant number was used as a random effect variable to control for variation in magnitude of image evoked changes across participants. Using this strategy, we were able to examine the effect of image valence on oscillatory dynamics across participants and contacts despite potential individual differences. To estimate task-related changes in bandpower, the total power within a band of interest was calculated by taking the sum of the power of frequencies that make up the band of interest (i.e., at a single time step, 5–9 Hz power was calculated as the sum of power at frequencies 5–9 Hz). The total bandpower over time was then obtained as a vector, which included data from the entire experimental session, across conditions. To perform normalization, the mean and standard deviation were calculated based on the bandpower of the entire signal throughout the experimental session (Eq. 1). This normalization approach was used to better reflect changes that were not as susceptible to increased trial-to-trial pre-stimulus variability and to enable comparisons with results from experimental tasks that did not contain trial-related baseline periods (e.g., script-driven-imagery task).

$$\frac{\text{Raw power} - \mu_{\text{raw power}}}{\sigma_{\text{raw power}}} \qquad (1)$$

This procedure was repeated for each channel similar to previous studies[22]. These LMMs were implemented using the SPSS statistical package (IBM Corp. Released 2020. IBM SPSS Statistics for Windows, Armonk, NY: IBM Corp).

To assess significance of differences in bandpower between traumatic and pleasant audio scripts and between self-reported symptom exacerbations and scheduled recordings during asymptomatic periods, the aforementioned LMM strategy was employed and traumatic/pleasant and symptomatic/asymptomatic were used as type variables, respectively. Given the length of scheduled and magnet swipe recordings, samples were divided into 60 s segments.

To assess significance of differences in bandpower between negative and positive/neutral images pre- and post-stimulation, stimulus evoked bandpower changes pre- and post-stimulation were normalized relative to one another (for each participant) and fed into the LMM with random effects of contacts and fixed effects of session and type (negative versus positive/neutral).

To assess significant differences between negative and positive/neutral images across frequency bands in TR-PTSD and non-TR-PTSD groups an LMM was performed at each time step.

To aid the reader's ability to interpret pre-stimulation experimental findings during the Emotional Image and Script-Driven-Imagery Tasks, we approximated the effect size using the Cohen's d statistic (Eq. 2)[42]. This method was selected due to the lack of a standard approach for calculating effect size for linear mixed model analyses.

$$\frac{\mu_1 - \mu_2}{\sqrt{\frac{(n_1-1)\sigma_1^2 + (n_2-1)\sigma_2^2}{n_1+n_2-2}}} \qquad (2)$$

### Prevalence of low-frequency oscillations during periods of detections vs. non-detections

To determine the prevalence of low frequency oscillations during device detection events that triggered stimulation, we calculated power

spectral density (PSD) of iEEG data (individually for both participants) collected during periods of detection and non-detection. To prevent stimulation-related artifact from affecting analyses, we collected this activity while stimulation was turned off (and device detection was still on) during the Script-Driven-Imagery Task (Pre-Stim and Post-Stim 1-3). The Script-Driven Imagery Task was ideal for this analysis as it provided long periods of stored data with few task events. Following identification of detection and non-detection periods using timestamps of Trigger 1 (Supplementary Table 5) onset and duration, oscillatory activity was detected using the BOSC method in MATLAB (*Electrophysiological Analyses*). For epochs of 3 s before each detection, mean PSD was calculated based on time frequency heat maps for each epoch. Non-detection episodes were extracted from the same recordings after removing the detection epochs (+1 s before and after) and mean PSDs were calculated in the same way. Detection vs. non-detection epochs were combined irrespective of task conditions. Separately for both participants, mean PSD in the theta frequency range (5–9 Hz) was compared during detection vs. non-detection periods by normalizing the 5–9 Hz power during detection events with the mean and standard deviation of 5–9 Hz power during non-detection events (from the same sensing channel and participant) and performing a one-sample $t$ test.

### Differences in daily stimulations

To assess significant differences in the amount of stimulation therapies that TR-PTSD 1 and TR-PTSD 2 received, a Kruskal-Wallis test was performed for each subject to test for a main effect of stimulation period on therapy counts (TR-PTSD 1: $X^2(2) = 35.76$, $p < 0.001$; TR-PTSD 2: $X^2(2) = 69.05$, $p < 0.001$). Wilcoxon rank sum tests with Bonferroni multiple comparison correction were subsequently performed to determine if the stimulation received during Post-Stim 1 was significantly different than Post-Stim 2 and Post-Stim 3 in each subject (TR-PTSD 1 Post-Stim 1 vs. Post-Stim 2: $W = 468.5$, $p < 0.001$; TR-PTSD 1 Post-Stim 1 vs. Post-Stim 3: $W = 377$, $p < 0.001$; TR-PTSD 2 Post-Stim 1 vs. Post-Stim 2: $W = 630$, $p < 0.001$; TR-PTSD 2 Post-Stim 1 vs. Post-Stim 3: $W = 632$, $p < 0.001$). All tests were implemented using MATLAB.

### Reporting summary

Further information on research design is available in the Nature Portfolio Reporting Summary linked to this article.

## Data availability

Data are available upon request. Non-TR-PTSD iEEG and imaging data will be uploaded to the National Institute of Mental Health Data Archive (NDA) after completion of grant R01MH124761. TR-PTSD iEEG and imaging data will be uploaded to the Data Archive BRAIN Initiative repository following completion of grant UH3NS107673. Source data are provided with this paper.

## Code availability

The custom computer code used to generate our results are available from the corresponding authors upon request.

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

## Acknowledgements

This work was supported by National Institutes of Health awards R01MH124761 (to N.S. and J.P.L.), UH3NS107673 (to J.P.L.), F31MH127922 (to J.G.), R00 MH106649 and R01 MH119089 (to A.A.), and R01MH120194 (to C.S.I). N.S. was also supported by the McKnight Foundation (Technological Innovations in Neuroscience Award), Ruth and Raymond Stotter Endowment, and the Keck Foundation (UCLA DGSOM Junior Faculty Award). C.S.I. was also supported by National Science Foundation award 2124252. The authors thank all members of the Suthana lab for useful discussions, and all participants for taking part in the study. We also thank Josue Avecillas-Chasin, Sabrina Levy Maoz, Lorenzo Bonacini, Andrew Leuchter, Brazil Bartholomew, Virginia Janovsky, Andy Lin, and Jonny Baham for helpful discussions and/or general assistance.

## Author contributions

J.P.L., J.G., J.S., R.K. and N.S. conceived of the study. J.P.L., J.G., J.S., M.S., Z.M.A., M.V., C.S.I., S.H., U.T., R.K., M.V., N.H., S.K., and J.C. contributed to data collection. J.G., Z.A., U.T., and D.V. programmed the Emotional Image Task. J.S. and R.K. created the scripts and audio recordings for the Script-Driven Imagery Task. U.T. and M.V. programmed the Script-Driven Imagery Task. J.G., J.S., M.S., Z.A., M.V., J.K., V.R.R., A.A., M.C., M.F., and N.S. conceptualized the data analytical approach. J.G., J.S., J.P.L., and N.S. performed electrode localization procedures. J.G., M.V., and J.S. performed data analyses. J.P.L, J.G., J.S., M.S., M.V., C.I., A.A., J.K., V.R.R., M.F., M.C., and N.S. supervised data analyses procedures and interpretation of data. J.P.L., A.B., J.S., R.K., S.K., and J.C. performed the clinical care aspects of the study. J.P.L., R.K., N.H., and M.V. contributed to programming of the RNS. A.B. and J.P.L. performed surgical procedures. J.P.L, J.G., J.S., and N.S. wrote the first draft of the manuscript and all authors contributed to the writing and revision of the manuscript. J.P.L. and N.S. supervised all aspects of the study.

## Competing interests

V.R.R. has served as a paid consultant for NeuroPace but declares no targeted funding from NeuroPace for this study. M.V. is an employee of NeuroPace, Inc., Mountain View. The remaining authors declare no competing interests.

## Additional information

[1]Department of Psychiatry and Biobehavioral Sciences, Jane and Terry Semel Institute for Neuroscience and Human Behavior, University of California, Los Angeles, CA, USA. [2]Medical Scientist Training Program, University of California, Los Angeles, CA, USA. [3]Research and Development Service; Department of Veterans Affairs Greater Los Angeles Healthcare System, Los Angeles, CA, USA. [4]Department of Neurosurgery, University of California, Los Angeles, CA, USA. [5]Department of Electrical and Computer Engineering, University of California, Los Angeles, CA, USA. [6]Department of Psychology, University of Utah, Salt Lake City, UT, USA. [7]Department of Psychology, University of California, Los Angeles, CA, USA. [8]Weill Institute for Neurosciences, University of California, San Francisco, CA, USA. [9]Neurology Service; Department of Veterans Affairs Greater Los Angeles Healthcare System, Los Angeles, CA, USA. [10]Department of Neurology, University of California, Los Angeles, CA, USA. [11]NeuroPace Inc., Mountain View, CA, USA. [12]Phoenix Research Consulting LLC, Gilbert, AZ, USA. [13]Psychiatry and Mental Health Service; Department of Veterans Affairs Greater Los Angeles Healthcare System, Los Angeles, CA, USA. [14]Department of Bioengineering, University of California, Los Angeles, CA, USA. [15]Neurosurgery Service; Department of Veterans Affairs Greater Los Angeles Healthcare System, Los Angeles, CA, USA. [16]These authors contributed equally: Jay L. Gill, Julia A. Schneiders. [17]These authors jointly supervised this work: Nanthia Suthana, Jean-Philippe Langevin. ✉e-mail: nsuthana@mednet.ucla.edu; jlangevin@mednet.ucla.edu

