## [Peer Review File · Nature Communications]

A pilot study of closed-loop neuromodulation for treatment-resistant post-traumatic stress disorderEditorial Note: This manuscript has been previously reviewed at another journal that is not operating a transparent peer review scheme. This document only contains reviewer comments and rebuttal letters for versions considered at Nature Communications.

REVIEWERS' COMMENTS

Reviewer #1 (Remarks to the Author):

The authors have done a great job of answering all my queries. However, I have a few more suggestions to improve the revised manuscript.

To detect the significant clusters in the plot shown in Figure 2A cluster permutation tests were performed. If yes need to be described in the methods section.

In addition, the effect size for the comparisons in Figures 2 B and 2C would be a good indicator for the reader.

The reason for estimating the positive norm. pow difference and negative norm. pow difference is not clear to me instead of a standard approach of selecting a window of -2 to -1 second to look at trial-based relative power differences.

From a future perspective point of view would be good to have skin conductance response signals from these subjects would be a good indicator for emotional responses.

Reviewer #2 (Remarks to the Author):

This version of the article is certainly improved.

Minor Comments:

-Lines 130-135- It would facilitate the reading if the authors provided % reduction in CAPS scores for the different timepoints along the text. Just stating that results were significance is not enough. Looking at the Figure, it seems that patient 2 had less than 30% improvement.

-One potential reason for the more pronounced effect of stimulation over time would be the development of brain plasticity. This should be discussed.

Dear Editor and Reviewers,

We are pleased to learn that the reviewers found our revised manuscript improved, their questions answered, and had only minor comments to strengthen the manuscript further. We have incorporated each of their suggestions into the final version and outline these changes below.

Reviewer Comments:

Reviewer 1 [R1]: *The authors have done a great job of answering all my queries. However, I have a few more suggestions to improve the revised manuscript.*

Response: We thank the reviewer for their time and effort to review our manuscript, and for their valuable feedback. Please find our responses to each comment below.

Comment 1 [C1]: *To detect the significant clusters in the plot shown in Figure 2A cluster permutation tests were performed. If yes need to be described in the methods section.*

Response: We thank the reviewer for noticing the missing details related to the cluster-based permutation test and have now updated the methods section to include the following description (pg. 21):

“Significant temporal and frequency ranges were detected using cluster-based permutation testing. During this procedure, a null time-frequency power spectrogram was created by shuffling condition labels (negative vs. positive/neutral). An LMM was performed at each time-frequency power pair and a p value of < 0.05 was used to determine significance. Clusters (consecutive time-frequency pairs of significance) were then detected, and the size of the largest cluster was saved. This procedure was then repeated 1000 times. Cluster sizes derived from the observed (unshuffled data) that were larger than the 99th percentile of the shuffled distribution were labeled as significant.”

C2: *In addition, the effect size for the comparisons in Figures 2B and 2C would be a good indicator for the reader.*

Response: Due to the lack of a standard procedure for calculating effect size for linear mixed model analyses, we approximate effect size using the Cohen’s d statistic to aid the reader in interpreting results. We include the following in the manuscript (pg. 22):

“To aid the reader’s ability to interpret pre-stimulation experimental findings during the emotional image and script-driven-imagery tasks, we approximated the effect size using the Cohen’s d statistic (equation 2)¹. This method was selected due to the lack of a standard approach for calculating effect size for linear mixed model analyses.

$$\frac{\mu_1 - \mu_2}{\sqrt{\frac{(n_1-1)\sigma_1^2 + (n_2-1)\sigma_2^2}{n_1 + n_2 - 2}}} \quad (\text{equation 2})$$

C3: *The reason for estimating the positive norm. pow difference and negative norm. pow difference is not clear to me instead of a standard approach of selecting a window of -2 to -1 second to look at trial-based relative power differences.*

Response: We thank the reviewer for pointing out this detail related to our methods selection process. As the reviewer points out, the mean and standard deviation statistics used for data normalization (equation 1) were calculated from the entire trace (within a participant and within a single electrode) rather than a 1 second pre-stimulus window from the corresponding trial. However, in analyses not reported in the paper, we used both strategies (baseline vs. trace normalization) and found that they yielded identical

trends. However, trace normalization produced more stable stimulus evoked responses (likely due to the larger variability across trial-related pre-stimulus periods). Furthermore, trace normalization enabled the use of an identical approach across behavioral tasks since the exposure task did not have baseline pre-stimulus periods that could be used for normalization. To clarify the selection of our normalization approach we have added the following description to the methods section (pg. 22).

“To perform normalization, the mean and standard deviation were calculated based on the bandpower of the entire signal throughout the experimental session (equation 1). This normalization approach was used to better reflect changes that were not as susceptible to increased trial-to-trial pre-stimulus baseline variability and to enable comparisons with results from experimental tasks that did not contain trial-related baseline periods (e.g., script-driven-imagery task).”

$$\frac{\text{Raw power} - \mu_{\text{raw power}}}{\sigma_{\text{raw power}}} \quad (\text{equation 1})$$

C4: *From a future perspective point of view would be good to have skin conductance response signals from these subjects would be a good indicator for emotional responses.*

Response: We agree that skin conductance response measures would be a valuable addition to the study. Unfortunately, skin conductance response recordings of non-TR-PTSD participants were not collected. We have added a statement to our discussion section (pg. 9) suggesting that future studies should collect skin conductance response signals and that these recordings will be useful for further characterizing the relationship between neural activity and emotional responses.

“Future studies are needed to determine if heightened amygdala theta bandpower during negative images reflects pathological changes in amygdala circuitry, if these changes are specific to PTSD and/or other anxiety-related disorders, and if these effects are accompanied by changes in other physiological measures such as skin conductance response.”

Reviewer #2:

R2: *This version of the article is certainly improved.*

Response: We thank the reviewer for their time and effort to review our manuscript, and for their positive feedback. Please find our responses to each comment below.

C1: *-Lines 130-135- It would facilitate the reading if the authors provided % reduction in CAPS scores for the different timepoints along the text. Just stating that results were significance is not enough. Looking at the Figure, it seems that patient 2 had less than 30% improvement.*

-One potential reason for the more pronounced effect of stimulation over time would be the development of brain plasticity. This should be discussed.

Response: We agree with the reviewer that providing % symptom improvement in CAPS-5 and also PCL-5 scores gives a more comprehensible understanding of the degree of symptom improvement for both of our participants. Therefore, we calculated % symptom improvement on the CAPS-5 and on the PCL-5 for each participant by using the following formula: % symptom improvement = [(pre-stimulation baseline score (mean of baseline 1, baseline 2, month 1) – score of each month after stimulation onset (post-stim)) / pre-stimulation baseline score] x 100. We added these details to the methods section (pg. 6) and the results section of the manuscript accordingly:

“TR-PTSD 1 exhibited significant CAPS-5 reductions² (within-person change scores ≥ 13) during all clinical assessments following stimulation onset (month 2: 50.59% symptom improvement, month 3: 54.12%, month 4: 77.06%, month 5: 70.00%, month 6: 57.65%, month 7: 52.35%, month

8: 66.47%, month 9: 82.35%, month 10: 73.53%, month 11: 70.00%, month 12: 87.65%; (Fig. 3a, Suppl. Fig. 7a, 7c; subtests Suppl. Fig. 7e), while meaningful clinical improvements were observed in TR-PTSD 2 during months 4 (21.05% symptom improvement), 7 (32.11%), 8 (32.11%), 11 (40.00%) and 12 (36.84%) (Fig. 3b, Suppl. Fig. 7b, 7d; subtests Suppl. Fig. 7f). Subjectively-rated PTSD symptom changes as measured by PCL-5 were clinically reliable² (within-person change scores ≥ 15) during months 3-5 (month 3: 32.60% symptom improvement, month 4: 59.03% , month 5: 61.67%) and 7-12 (month 7: 29.96%, month 8: 47.14%, month 9: 57.71%. month 10: 55.07%, month 11: 56.39%, month 12: 63.00%) for TR-PTSD 1 and during months 11-12 (month 11: 34.03% symptom improvement, month 12: 27.75%) for TR-PTSD 2 (Suppl. Fig. 7a-d; subtests Suppl. Fig. 7g-h).”

“We calculated % symptom improvement on the CAPS-5 and on the PCL-5 for each participant by using the following formula: % symptom improvement = [(pre-stimulation baseline score (mean of baseline 1, baseline 2, month 1) – score of each month after stimulation onset (post-stim)) / pre-stimulation baseline score] x 100.”

To illustrate this further, we added two additional panels to Suppl. Fig. 7 (panel c and d) showing % symptom improvement on the CAPS-5 and PCL-5 for pre-stimulation and for each month after stimulation onset:

Supplementary Figure 7. CAPS-5 and PCL-5 subtest scores over time. Changes in CAPS-5 scores (gray) and PCL-5 scores (cyan) for TR-PTSD 1 (a) and TR-PTSD 2 (b) during Pre-Stim periods (baseline

1, 2, and month 1) and Post-Stim periods (months 2-12). Gray/cyan * = reliable change in CAPS-5 or PCL-5 scores, respectively, according to threshold for sample 1 reported in Marx et al.² relative to pre-stimulation baseline assessments (mean of baseline 1, 2, and month 1). **c-d**: % symptom improvement on CAPS-5 (gray) and PCL-5 (cyan) during Pre-Stim periods (baseline 1, 2, and month1) and Post-Stim periods (months 2-12). Gray/cyan * = reliable change in CAPS-5 or PCL-5 scores, respectively, according to threshold for sample 1 reported in Marx et al.² relative to pre-stimulation baseline assessments (mean of baseline 1, 2, and month 1).

Moreover, as suggest by the reviewer we have added a discussion (pg. 10) of neuroplasticity as a potential reason for the more pronounced effect of stimulation over time:

“Moreover, we have shown that amygdala closed-loop neurostimulation leads to a decline in both PTSD symptomatology and amygdala reactivity to aversive stimuli, which may be due to neuroplasticity-related changes over time.”

References

1. Cohen, J. *Statistical power analysis for the behavioral sciences*. (L. Erlbaum Associates, 1988).
2. Marx, B. P. *et al.* Reliable and clinically significant change in the clinician-administered PTSD Scale for DSM-5 and PTSD Checklist for DSM-5 among male veterans. *Psychological Assessment* **34**, 197–203 (2022).